# Manipulating hydrogen bond dissociation rates and mechanisms in water dimer through vibrational strong coupling

Qi Yu[1,2] ✉ & Joel M. Bowman [2]

The vibrational strong coupling (VSC) between molecular vibrations and cavity photon modes has recently emerged as a promising tool for influencing chemical reactivities. Despite numerous experimental and theoretical efforts, the underlying mechanism of VSC effects remains elusive. In this study, we combine state-of-art quantum cavity vibrational self-consistent field/configuration interaction theory (cav-VSCF/VCI), quasi-classical trajectory method, along with the quantum-chemical CCSD(T)-level machine learning potential, to simulate the hydrogen bond dissociation dynamics of water dimer under VSC. We observe that manipulating the light-matter coupling strength and cavity frequencies can either inhibit or accelerate the dissociation rate. Furthermore, we discover that the cavity surprisingly modifies the vibrational dissociation channels, with a pathway involving both water fragments in their ground vibrational states becoming the major channel, which is a minor one when the water dimer is outside the cavity. We elucidate the mechanisms behind these effects by investigating the critical role of the optical cavity in modifying the intramolecular and intermolecular coupling patterns. While our work focuses on single water dimer system, it provides direct and statistically significant evidence of VSC effects on molecular reaction dynamics.

Strong light–matter interactions between molecules and the electromagnetic field of an optical cavity have garnered significant interest in the fields of chemistry and materials, as they provide new opportunities to modify chemical reactivity and selectivity[1-7]. When the infrared (IR) cavity mode is strongly coupled to specific vibrational excitations, known as vibrational strong coupling (VSC), the molecular vibrational polaritons are formed and shown to potentially alter the chemical reaction and energy transfer pathways[8-10]. A series of seminal experimental work have been conducted to monitor reaction pathways and intramolecular vibrational energy redistribution (IVR) process by detecting the IR signature of molecular vibrational polaritons with linear IR spectroscopy and nonlinear two-dimensional IR spectroscopy (2D-IR)[8-14].

Despite numerous theoretical efforts trying to reach consensus with experimental measurements, the mechanism by which vibrational strong coupling controls chemical reactions remains obscure[15-26]. This is mainly attributed to the complexity of molecule-cavity systems and the lack of promising theoretical tools to disentangle the reaction mechanisms. For example, some initial attempts involved investigating chemical reactions under VSC with transition state theory, using simple analytical expressions for the potential energy and dipole moment[22,23,27]. Recently, Wang and coworkers performed extensive trajectory-based dynamics simulations of unimolecular dissociation reactions inside an optical cavity and demonstrate how VSC interferes with the IVR process and alters the dissociation reaction rate[24]. However, this work was limited by the accuracy of the simple Morse potential and dipole moment functions, which are not generalizable to realistic molecular systems. Additionally, the rotational degrees of freedom were ignored in the simulations, which are important from the perspective of reaction dynamics. More recently, Schafer and

[1]Department of Chemistry, Yale University, New Haven, CT 06520, USA. [2]Department of Chemistry, Emory University and Cherry L. Emerson Center for Scientific Computation, Atlanta, GA 30322, USA. ✉e-mail: qyu28@emory.edu

coworkers applied the quantum-electrodynamical density functional theory (QED-DFT) approach in the ab initio molecular dynamics (AIMD) simulation of the deprotection reaction of 1-phenyl-2-trimethylsilylacetylene (PTA)[26]. Qualitative agreement with experiments was successfully achieved in terms of reaction rate inhibition. Although many fundamental insights were obtained from the AIMD simulation, the high computational cost of the QED-DFT approach limits the number of trajectories as required for statistical significance. The accuracy of the QED-DFT approach is also limited due to similar issues in the conventional DFT method. More accurate methods, such as the QED-coupled cluster (QED-CC) method[28,29], are more computationally expensive and becomes prohibitive in extensive reaction dynamics simulations. Such limitations pose challenges in theoretically understanding the microscopic mechanism of how VSC controls chemical reactions, using state-of-art quantum approaches, such as multiconfigurational time-dependent Hartree method (MCTDH)[30,31] and vibrational self-consistent field/virtual state configuration interaction method (VSCF/VCI)[32,33].

As an alternative to the simple models or AIMD approach, high-dimensional, ab initio-based machine learning (ML) potential energy and dipole moment surfaces (PES/DMS) are natural choices for achieving both high accuracy and computational efficiency in both quantum and classical dynamics simulations[34,35]. With the aid of ML PES and DMS, extensive quasi-classical trajectory (QCT) calculations can be performed to realize full-dimensional reaction dynamics simulations. The QCT approach has been widely applied in the field of reaction dynamics due to its simplicity and capability of providing rich information of the dynamical process. Through choosing initial coordinates and velocities of the reactants, which correspond to the quantum mechanical (ro-)vibrational energy levels, the QCT approach enables vibrational mode specific control of the reaction dynamics simulation.

In this work, we combined our recently developed quantum cavity vibrational self-consistent field/virtual state configuration interaction (cav-VSCF/VCI) method[36] and QCT approach to investigate the effect of vibrational strong coupling on hydrogen-bond dissociation dynamics of the water dimer $(H_2O)_2$, using our recently developed ML PES/DMS, q-AQUA, at the CCSD(T)/CBS level of accuracy[37]. As the smallest water cluster, the dimer serves as a benchmark for both experiment and theory providing valuable insights into hydrogen bond breaking and reformation in complex water networks from the gas phase to condensed phase[38–41]. Previous experimental and theoretical studies consistently found the predominant excitation of the fragment bending mode upon the hydrogen bond breaking[42]. In addition to this dominant dissociation channel, a very minor channel was also observed where the two water fragments are in their vibrational ground states and highly rotationally excited.

We first confirm the previous experimental and theoretical findings concerning vibrational spectra and dissociation dynamics information when $(H_2O)_2$ is outside the cavity. When $(H_2O)_2$ is inside the cavity, through QCT simulations, we investigated how the hydrogen bond dissociation rate can be either inhibited or accelerated by manipulating the light–matter coupling strength and cavity frequency. Additionally, we analyzed the vibrational product populations of the dissociated fragments and explored how the vibrational strong coupling modifies the dissociation channels. The underlying mechanism of the cavity-modified hydrogen bond dissociation dynamics is discussed through the QCT simulations and vibrational spectral computations.

## Results
### Polaritonic IR spectra and mode-coupling patterns
The water dimer is put in an optical cavity, as shown in Fig. 1a, and a single cavity mode is considered with its polarization direction along

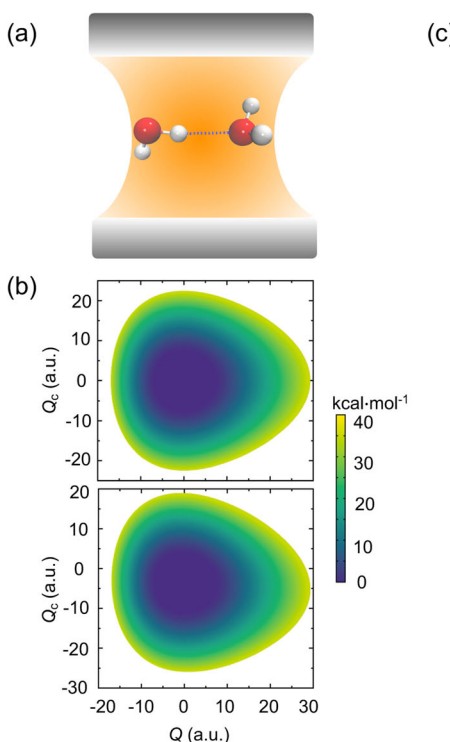

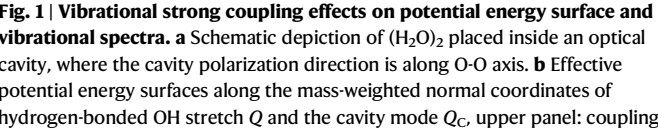

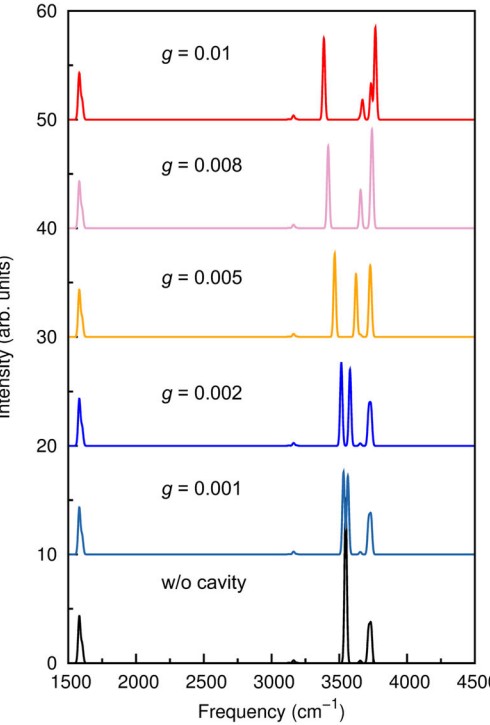

**Fig. 1 | Vibrational strong coupling effects on potential energy surface and vibrational spectra. a** Schematic depiction of $(H_2O)_2$ placed inside an optical cavity, where the cavity polarization direction is along O-O axis. **b** Effective potential energy surfaces along the mass-weighted normal coordinates of hydrogen-bonded OH stretch $Q$ and the cavity mode $Q_C$, upper panel: coupling strength $g = 0.0$, lower panel: coupling strength $g = 0.005$. **c** Infrared spectra of $(H_2O)_2$ with different light–matter coupling factors $g$. For **b**, **c**, the single cavity mode has a frequency of 3547 cm$^{-1}$, which is the fundamental hydrogen-bonded OH stretch mode frequency of the water dimer outside the cavity.

O-O axis of the water dimer. The coupling strength between the cavity and $(H_2O)_2$ is controlled by the light–matter coupling factor $g$, as defined in Eq. (6) in the "Methods" section. It should be noted that in the vibrational strong coupling experiments, a macroscopic number of molecules, $N_{mol}$ (i.e., $10^{10}$), are collectively coupled to the cavity and the resulting Rabi splitting is proportional to $\sqrt{N_{mol}/\tilde{V}}$. In the current work, the light–matter coupling is directly between the cavity and the water dimer. Thus to compensate for the lack of $N_{mol}$ molecules in our $(H_2O)_2$-cavity system, the coupling strength factor $g$ throughout this work indicates a much smaller cavity volume $\tilde{V}$ than the experimental set-up. For example, with cavity frequency $\omega = 3547$ cm$^{-1}$ and coupling strength factor $g = 0.005$ a.u., the effective cavity volume $\tilde{V}$ calculated from Eq. (6) is 0.6 nm$^3$ which is within the picocavity range. We also emphasize that the single-molecule simulations in this work serves as a good example to investigate how the optical cavity affects the vibrational mode couplings and chemical reaction dynamics. To bridge the gap between single-molecule simulation and realistic experimental conditions in collective coupling regime, the many-molecule system will need to be considered and this is subject to our future work using extensions of the approaches as described in the "Methods" section.

In Fig. 1b, we show the effective potential energy surface along the hydrogen-bonded OH stretch (HB stretch) and the cavity mode when the coupling strength factor $g$ is 0.0 or 0.005. The former case is equivalent to the situation that $(H_2O)_2$ is outside the cavity. As seen from the upper panel of Fig. 1b, the potential along the HB stretch mode is asymmetric and strongly anharmonic while the potential along the cavity mode is essentially the harmonic potential. When $(H_2O)_2$ is inside the cavity, the potential energy surface, shown in the lower panel of Fig. 1b, is significantly different from the uncoupled one, in terms of the position of minimum energy and curvature of the PES. Such change of the PES indicates that both spectral and dynamical properties of the $(H_2O)_2$-cavity system may be altered by vibrational strong coupling.

Before presenting the calculated IR spectra of the $(H_2O)_2$-cavity system, it is crucial to emphasize the accuracy of our PES and DMS for $(H_2O)_2$. Supplementary Table 1 gives harmonic frequencies and intensities of 12 normal modes of $(H_2O)_2$ at its global minimum structure using our PES/DMS and ab initio methods. As seen, the harmonic

frequencies predicted from our PES outperform the MP2/aVTZ results and agree very well with the CCSD(T)-F12b/aVTZ benchmark calculations, with differences of no more than 1 cm$^{-1}$ for high-frequency modes and 5 cm$^{-1}$ for several low-frequency modes. The double harmonic intensities of all normal modes also match well with the MP2/aVTZ calculations. Additional studies of the accuracy of our PES/DMS are provided in previous papers[37,43]. It should also be noted that, to obtain frequencies of polaritonic states, the ubiquitous Jaynes-Cummings model[44,45] could be applied by solving simplified polariton Hamiltonian matrix. However, it has been shown that the Jaynes-Cummings model is not accurate in describing rovibrational transitions[46]. Such method is also limited to two-level systems and ignores the permanent dipole and dipole self-energy[47]. Our cav-VSCF/VCI approach applies a rigorous Hamiltonian and could accurately describe multi-level couplings among molecular vibrations and cavity modes[37].

In Fig. 1c, we show the cav-VSCF/VCI spectra of the $(H_2O)_2$-cavity system by varying the light–matter coupling factor $g$. First, when $(H_2O)_2$ is outside the cavity, all OH stretches experience a redshift of about 200 cm$^{-1}$ compared to their harmonic frequencies. One prominent peak at 3547 cm$^{-1}$ is observed which corresponds to the fundamental band of the HB stretch. This is in good agreement with a recent experimental measurement of 3549 cm$^{-1}$ using vacuum ultraviolet free electron laser (VUV-FEL)[48]. Another intense spectral peak at around 3720 cm$^{-1}$ results from the free OH stretch of the donor water (3716 cm$^{-1}$) and the asymmetric OH stretch of the acceptor water (3734 cm$^{-1}$). The symmetric OH stretch of the acceptor water at 3655 cm$^{-1}$ has low IR intensity and is barely visible in Fig. 1c. The spectral peak around 1600 cm$^{-1}$ comes from the fundamental band of the bending modes of the donor and acceptor waters. Again, all these band positions agree well with previous experiments[48,49]. These observations further verify the accuracy of our PES/DMS and highlight the need for quantum dynamical approaches, such as VSCF/VCI, in treating strongly anharmonic molecular systems.

When $(H_2O)_2$ is placed in an optical cavity with cavity mode frequency as 3547 cm$^{-1}$ and polarization direction along O-O axis, the original HB stretch at 3547 cm$^{-1}$ splits into two intense peaks, corresponding to the lower polariton (LP) and upper polariton (UP). As shown in Table 1, both LP and UP states are dominated by

**Table 1 | Cavity vibrational self-consistent field/configuration interaction (cav-VSCF/VCI) state energies and leading vibrational configuration interaction (VCI) coefficient(s) under different values of the light–matter coupling factor $g$ with the cavity mode polarized along the O-O axis**

| Cavity mode frequency $\omega = 3547$ cm$^{-1}$ | | | | | | | | |
|---|---|---|---|---|---|---|---|---|
| **Coupling factor $g$** | **0.00** | | | **0.001** | | | **0.002** | | |
| Energy (cm$^{-1}$) | 3547 | 3547[a] | 3716 | 3531[b] | 3563[c] | 3717 | 3514[b] | 3579[c] | 3717 |
| VCI coeff (cavity) | / | 1.0 | / | 0.71 | 0.70 | 0.04 | 0.71 | 0.70 | 0.09 |
| VCI coeff (HB str) | 0.97 | / | 0.09 | 0.68 | 0.69 | 0.09 | 0.67 | 0.70 | 0.08 |
| VCI coeff (free str) | 0.16 | / | 0.99 | 0.09 | 0.05 | 0.99 | 0.11 | 0.04 | 0.98 |
| VCI coeff (sym str) | / | / | / | / | 0.05 | / | / | 0.06 | / |
| VCI coeff (bend overtone[d]) | 0.15 | / | 0.05 | 0.11 | 0.11 | 0.05 | 0.11 | 0.10 | 0.05 |
| **Coupling factor $g$** | **0.005** | | | **0.008** | | | **0.01** | | |
| Energy (cm$^{-1}$) | 3466[b] | 3623[c] | 3724 | 3418[b] | 3658[c] | 3743 | 3386[b] | 3671[c] | 3766 |
| VCI coeff (cavity) | 0.72 | 0.68 | 0.22 | 0.72 | 0.54 | 0.42 | 0.72 | 0.44 | 0.52 |
| VCI coeff (HB str) | 0.66 | 0.72 | 0.02 | 0.64 | 0.69 | 0.21 | 0.63 | 0.65 | 0.35 |
| VCI coeff (free str) | 0.15 | 0.16 | 0.96 | 0.18 | 0.43 | 0.87 | 0.20 | 0.59 | 0.77 |
| VCI coeff (sym str) | / | 0.05 | / | / | 0.16 | / | / | 0.18 | / |
| VCI coeff (bend overtone[d]) | 0.13 | 0.10 | 0.04 | 0.15 | 0.11 | 0.01 | 0.17 | 0.11 | <0.01 |

"/" indicates the VCI coefficients smaller than 0.005. "HB str", "free str", and "sym str" indicate the hydrogen bonded OH stretch, free OH stretch, and symmetric OH stretch respectively.
[a]Fundamental frequency of the cavity mode.
[b]Lower polariton state (LP).
[c]Upper polariton state (UP).
[d]Bend overtone of the donor water.

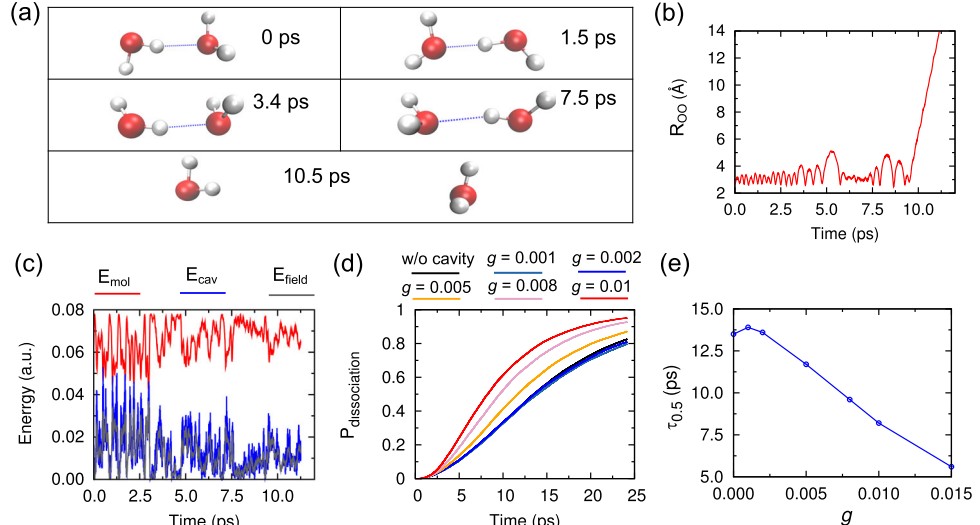

**Fig. 2 | Modification of dissociation rate from different light–matter coupling strengths. a** Selected frames labeled in picoseconds at different time in a representative quasi-classical trajectory (QCT) leading to dissociation. **b** O-O distance ($R_{OO}$) in $(H_2O)_2$ as a function of time in the representative QCT trajectory. **c** Total energy of the water dimer, cavity mode, and field ($E_{mol}, E_{cav}, E_{field}$), as a function of time in the representative QCT trajectory. **d** Percentage of dissociated trajectories as a function of time for each $(H_2O)_2$-cavity system with different light–matter coupling factors $g$. **e** Half-lifetime ($\tau_{0.5}$) of $(H_2O)_2(\nu_{OH}=1)$ for $(H_2O)_2$-cavity systems with different light–matter coupling factor $g$. ($\nu_{OH}=1$) indicates one quanta excitation of the hydrogen-bonded OH stretch. The cavity frequency is set as 3547 cm$^{-1}$ for **a**–**e** and the light–matter coupling factor $g$ is set as 0.005 for **a**–**c**.

contributions from the cavity mode and the HB stretch mode, indicating their strong couplings. Additionally, the LP state has a notable contribution from the free OH stretch and bend overtone of the donor water, while the UP state has additional involvement from the symmetric OH stretch of the acceptor water. The direct coupling between the acceptor water's symmetric OH stretch and donor water's free OH stretch or HB stretch is nearly neglectable when $(H_2O)_2$ is outside the cavity. However, it becomes possible when $(H_2O)_2$ is inside the cavity with the critical role of the cavity mode as a mediator. The involvement of donor water's free OH stretch is straightforward since its dipole moment vector is non-perpendicular to the cavity mode's polarization direction. As for the symmetric stretch, the UP state from HB stretch and cavity mode locates close to the symmetric OH stretch region, thus resonance with the symmetric OH stretch is realized. As seen in Fig. 1c and Table 1, the splitting between the LP and UP states, known as Rabi splitting, becomes greater with an increase in the light–matter coupling factor, $g$. The mixing between the HB stretch, cavity mode, free OH stretch, and symmetric OH stretch in the UP state becomes stronger with a higher light–matter coupling strength. This further suggests that the optical cavity can induce the coupling of different vibrational modes, leading to acceleration of intramolecular and intermolecular vibrational energy transfer. It is important to note that the bend overtone is also mixed with the HB stretch even when $(H_2O)_2$ is outside of the cavity. Similar to the stretching modes, the mixing with bend overtone is more significant as the light–matter coupling factor $g$ increases. As will be discussed in the dissociation dynamics, the coupling to the bend overtone can be important to the possible dissociation channel. It can also be observed in Fig. 1c that the Rabi splitting patterns are slightly asymmetric with increasing light–matter coupling strength. As seen in Table 1, the asymmetric intensities of LP and UP states are attributed to different amounts of molecular vibrational modes in the formation of hybrid polaritonic state (see corresponding VCI coefficients). It would also be interesting to test approximate methods, such as perturbation theory[50,51], to capture the asymmetric Rabi splitting features. Finally, in the current cav-VSCF/VCI calculations, only high-frequency vibrational modes in $(H_2O)_2$ were considered. To obtain more accurate IR spectra, the low-frequency modes should be included. As shown in Supplementary Table 2, the

preliminary results with the inclusion of three intermolecular low-frequency modes (O-O stretch, out-of-plane bend, and in-plane bend) indicate that the HB stretch can be significantly mixed with complex combination bands involving low-frequency modes and water bend. However, due to the difficulty in obtaining converged spectra when low-frequency modes are included[52], we decide to present the cav-VSCF/VCI results considering only high-frequency modes, which have been shown above to be accurate enough compared to experiments.

### Hydrogen-bond dissociation rate controlled by light–matter coupling strength

The quantum dynamical simulations of the IR spectra of $(H_2O)_2$-cavity system provide direct evidence of the origin of polariton states' special spectral signatures and promising predictions of cavity-mediated intramolecular/intermolecular vibrational energy transfer pathways. To explicitly figure out the role of light–matter interaction in altering the chemical reaction dynamics, reliable and extensive dynamics simulations are needed. Next, we present the QCT simulation results of the vibrational dissociation dynamics of $(H_2O)_2$ within an optical cavity. In Fig. 2a, snapshots of a representative dissociation trajectory are shown. As seen, as time evolves, the identity of the donor and acceptor water switches several times. At a certain time, the two water fragments start to move apart and the dissociated monomers are observed. Such observation agrees with the time evolution of O-O distance in Fig. 2b along the same trajectory. In Fig. 2c, we show the time evolution of the total energy of $(H_2O)_2$, cavity mode, and the field. As seen, there exists significant energy exchange between the molecule and cavity field, where the sum of $E_{mol}$ and $E_{field}$ is set as a constant along the microcanonical (NVE) trajectory. The energy of the cavity mode follows a similar oscillation trend as the field energy. Note, due to the well-known ZPE leak issue of the QCT approach, the rate and amplitude of energy exchange shown in Fig. 2c are overestimated. Despite this fact, the QCT approach is known to be able to provide enriching information on the dynamics.

Figure 2d shows the percentage of dissociated trajectories among 20,000 QCT trajectories as a function of time, for different initial conditions of light–matter coupling strength $g$. Figure 2e is the

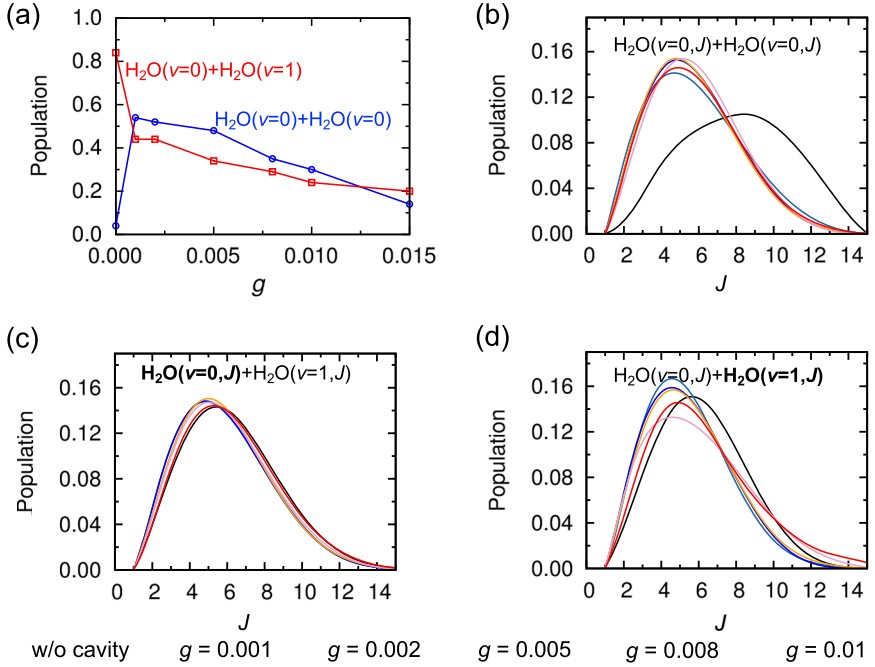

**Fig. 3 | Cavity-induced modification on dissociation channel. a** Vibrational product population for the dissociation of the water dimer inside an optical cavity with different light–matter coupling factors $g$. The channel of vibrational ground state products, (000)+(000), is indicated in blue and the channel of one bend-excited monomer fragment, (000)+(010), is indicated in red. **b–d** Rotational distributions of products with different coupling factors $g$. **b** for vibrational ground state products in (000)+(000) channel; **c**, **d** for vibrational ground state products and bend-excited products in (000)+(010) channel, respectively. $(n_1 n_2 n_3)$ indicates $n_1$, $n_2$, and $n_3$ excitations of the symmetric stretch, bend, and asymmetric stretch mode respectively. Both (000) and $(v=0,J)$ indicate the vibrational ground state of the dissociation product with $J$ as the rotational state. Both (010) and $(v=1,J)$ indicate the bending fundamental state of the dissociation product with $J$ as the rotational state.

corresponding estimations of the half-lifetime of $(H_2O)_2(v_{OH}=1)$. First, when $(H_2O)_2$ is outside the cavity, at around 25 ps about 83% of the trajectories dissociated and QCTs predict the half-lifetime of $(H_2O)_2(v_{OH}=1)$ as about 13.5 ps. This agrees well with previous theoretical investigations of $(H_2O)_2$ dissociation dynamics[42,53]. When $(H_2O)_2$ is inside the cavity with resonance frequency of 3547 cm[-1] and light–matter coupling factor $g$ of 0.002, the corresponding dissociation percentage curve (blue one) is very close to the case without cavity (black one) across the whole 25 ps period. The predicted half-lifetime is 13.6 ps which is almost identical to the prediction of the system without cavity. An intuitive feeling is that relatively weak light–matter coupling may not affect the dynamics at all. However, when the light–matter coupling factor $g$ is further decreased to 0.001, as indicated in the sky blue curve in Fig. 2d, the dissociation of $(H_2O)_2$ is slowed down although not dramatically. The corresponding half-lifetime is predicted as 13.9 ps which is non-negligibly longer than the 13.5 ps in the case without cavity. Thus, even when the light–matter coupling strength is fairly weak, the cavity will still affect the reaction dynamics, playing the role of reservoir for the energy and inhibiting the intramolecular/intermolecular vibrational energy transfer during the hydrogen-bond dissociation dynamics. This indicates the initial intuitive explanation is problematic for the case when the coupling factor $g$ is 0.002. Actually, the cavity field interacts with the molecule by both absorbing and releasing energy. At the same time, the cavity mode induces the coupling among various vibrational modes which could make intramolecular/intermolecular vibrational energy transfer more efficient. The above cavity-molecule energy exchange, as well as the acceleration of intermolecular/intramolecular vibrational energy transfer, can reach a dynamical balance and the final reaction rate is not affected by the existence of cavity, as shown in the case of coupling factor $g=0.002$. We will discuss this more in the following

section on the vibrational dissociation channels. When the light–matter coupling strength becomes weaker, the cavity field mainly takes the role of absorbing energies from the molecule and the vibrational energy transfer in $(H_2O)_2$ can not be efficiently improved. Then the final reaction rate is inhibited. When the coupling strength is increased, as seen in both Fig. 2d, e, the dissociation reaction gets accelerated significantly. For example, in the case of coupling factor $g$ as 0.01, at 25 ps, 95.2% of the trajectories dissociated and the half-lifetime of $(H_2O)_2(v_{OH}=1)$ is estimated as 8.2 ps only. For such cases, the frequent and intense energy exchange between molecule and cavity field weakens the role of the cavity as a reservoir for energy. Furthermore, as indicated in previous experiment and theory[42,53], when $(H_2O)_2$ is outside the cavity, the coupling to the dissociation coordinates is inefficient which requires sufficient time for the energy to redistribute among the available vibrational states and only restricted paths can lead eventually to dissociation. Now, under strong light–matter coupling, the cavity mode could induce couplings among $(H_2O)_2$'s vibrational modes, which will accelerate the vibrational energy redistribution and thus shorten the time needed for dissociation.

### Dissociation pathways altered by vibrational strong coupling

Besides the reaction rate, it is important to investigate how the cavity affects the chemical selectivity, such as the vibrational dissociation channels in $(H_2O)_2$. We analyzed the vibrational and rotational populations of water monomer products in dissociated trajectories. Figure 3(a)–(d) show these population distributions of different dissociation channels for $(H_2O)_2$-cavity systems with different light–matter coupling strengths. Again, let's first consider the case when $(H_2O)_2$ is outside the cavity. As shown in Fig. 3a, the dominant vibrational dissociation channel is (000)+(010) where (000) indicates the vibrational ground state and (010) is the first-excited bending state

**Table 2 | Vibrational product distributions for the dissociation of the water dimer for $(H_2O)_2$-cavity systems with different coupling strengths**

| $(H_2O)_2(v_{OH}=1) \rightarrow H_2O(n_1 n_2 n_3) + H_2O(n_1' n_2' n_3')$ | | | | | | | |
|---|---|---|---|---|---|---|---|
| | Light−matter coupling factor $g$ | | | | | | |
| $H_2O(n_1 n_2 n_3) + H_2O(n_1' n_2' n_3')$ | 0.0 | 0.001 | 0.002 | 0.005 | 0.008 | 0.01 | 0.015 |
| (000)+(000) | 0.04 | 0.54 | 0.52 | 0.48 | 0.35 | 0.30 | 0.14 |
| (000)+(010) | 0.84 | 0.44 | 0.44 | 0.34 | 0.29 | 0.24 | 0.20 |
| (000)+(100) | 0.02 | 0.00 | 0.01 | 0.05 | 0.08 | 0.04 | 0.05 |
| (000)+(001) | 0.00 | 0.00 | 0.00 | 0.02 | 0.04 | 0.03 | 0.03 |
| (000)+(020) | 0.06 | 0.01 | 0.02 | 0.06 | 0.11 | 0.11 | 0.09 |
| (010)+(010) | 0.03 | 0.00 | 0.01 | 0.04 | 0.03 | 0.04 | 0.05 |
| sum | 0.99 | 0.99 | 1.0 | 0.99 | 0.90 | 0.76 | 0.56 |

The cavity frequency is set as 3547 cm$^{-1}$. (000) indicates the vibrational ground state, (010) and (020) indicates the bending fundamental and overtone states, (100) indicates the first-excited symmetric stretch state, and (001) indicates the first-excited asymmetric stretch state. ($v_{OH}$ = 1) indicates one quanta excitation of the hydrogen-bonded OH stretch.

of the water molecule. This product channel accounts for 84% of the total population. The (000)+(000) channel, where both water products stay in the vibrational ground state, is a minor channel with only 4% population. As a result, their rotational distributions are significantly hotter than those of the (000)+(010) channel. This is verified in Fig. 3b. The rotational distribution is broad for the (000)+(000) channel with the peak around $J = 8$. For the (000)+(010) channel, the rotational distribution is less broad with the peak around $J = 4$. All these observations agree well with previous theoretical investigations[42,53] as well as experimental measurements[42]. Conventionally, the energy transfer process following the excitation of the HB stretch of $(H_2O)_2$ (3547 cm$^{-1}$) is closely related to the intramolecular bending motion. For example, the HB stretch couples strongly with certain complicated combination bands involving intramolecular water bend and intermolecular low-frequency modes (in-plane bend for example), and also directly coupled with water bend overtone. The energy transfer will naturally decay to one involving the excitation of the dissociation degree of freedom, which often leaves one quanta of bend excitation in a fragment. Further energy decay from (010) to (000) is energetically relatively inefficient. Thus the (000)+(000) is a minor channel when $(H_2O)_2$ is outside the cavity. Note, there are other minor channels with excited stretch mode or double-excited bend mode (see Table 2). These channels are energetically closed quantum mechanically after considering the ZPE and dissociation energy $D_0$[42]. Due to ZPE leakage of the QCT approach, these channels are still observed.

Now, let's move to the results of the vibrational dissociation channels when $(H_2O)_2$ is inside an optical cavity. In the last section, we show that when the light−matter coupling factor $g$ is 0.002, the rate of vibrational dissociation of $(H_2O)_2$ remains almost unchanged compared to the case without cavity. However, as shown in Fig. 3 and Table 2, the vibrational product populations are significantly altered, with the major dissociation channel shifting from (000)+(010) to (000)+(000), which now accounts for 52% of the population. This striking change of vibrationally hydrogen bond dissociation channels of $(H_2O)_2$ highlights the critical role of the cavity in the chemical reactions. The significant modification of the dissociation channel by the cavity can be attributed to the novel couplings among HB stretch, bend overtone, and the cavity mode. After the bend overtone state (020) is excited through IVR process, direct decaying to the bend fundamental state (010) is not efficient due to their small vibrational couplings as indicated in Supplementary Table 3. Instead, the bend overtone can directly decay to the ground state (000) with energy absorbed by the cavity field. This is realized through HB stretch-mediated coupling between bend overtone and cavity mode. As shown in Supplementary Fig. 2 and Supplementary Table 3 where we performed cav-VSCF/VCI calculation with and without the inclusion of HB stretch, the coupling between bend

overtone and cavity mode becomes significant only when the HB stretch is included in the calculation. The fact that the HB stretch strongly couples with the cavity mode provides another new vibrational dissociation channel with the existence of VSC. Instead of transferring to the bend overtone and decaying to one quanta of bending excitation in a fragment, the vibrational energy of the excited HB stretch can be directly absorbed by the cavity field due to their strong coupling. The corresponding dissociation pathway involves energy transfers to one involving excitation of the dissociative degree of freedom, with vibrational ground state in a fragment and excess energies stored in the cavity field. This dissociation pathway is energetically efficient, resulting in the relative dominant (000)+(000) products. For both cases, the final products will not be rotational highly excited since the excess energies are absorbed by the cavity field. Direct evidence can be seen in the blue line of Fig. 3b. Instead of highly excited rotational states, the (000)+(000) products have apparently low rotational energy than those in the case without cavity. Again, this is because the excess energies are absorbed by the cavity field. As to the rotational states of (000)+(010) products, they do not differ much from the case without cavity.

Another important question is how the vibrational dissociation channels in $(H_2O)_2$-cavity system changes under different light−matter coupling strengths. As shown in Fig. 3a and Table 2, when the light−matter coupling factor $g$ decreases to 0.001, the population of (000)+(000) channel increases to 54%, reflecting the cavity's increasing role as an energy reservoir. This agrees with the findings in the dissociation reaction rate with $g = 0.001$, where the reaction is slightly inhibited. Conversely, as the light−matter coupling factor $g$ increases, the populations of both (000)+(000) and (000)+(010) channels decrease. As discussed in the section of IR spectra and also in Table 1, when the light−matter coupling strength increases, the couplings among cavity, HB stretch, free OH stretch, symmetric OH stretch, and the bend overtone becomes more significant. Additionally, when the intermolecular low-frequency modes (i.e., O-O stretch) are considered, as shown in Supplementary Table 2, vibrational modes such as bend overtone and free OH stretch are further mixed with the cavity mode and HB stretch with larger VCI coefficients. Thus, the intramolecular and intermolecular energy transfer involving these modes becomes more efficient. The energy can decay to the excitation of dissociation degree of freedom which is related with the low-frequency modes, leaving the fragment in its bend overtone or OH stretch excited state. With the contribution from the cavity field, such channels become energetically allowed quantum mechanically. For the distributions of rotational states of (000)+(000) and (000)+(010) channels, as seen from Fig. 3(b)−(d), they do not change significantly when the light−matter coupling

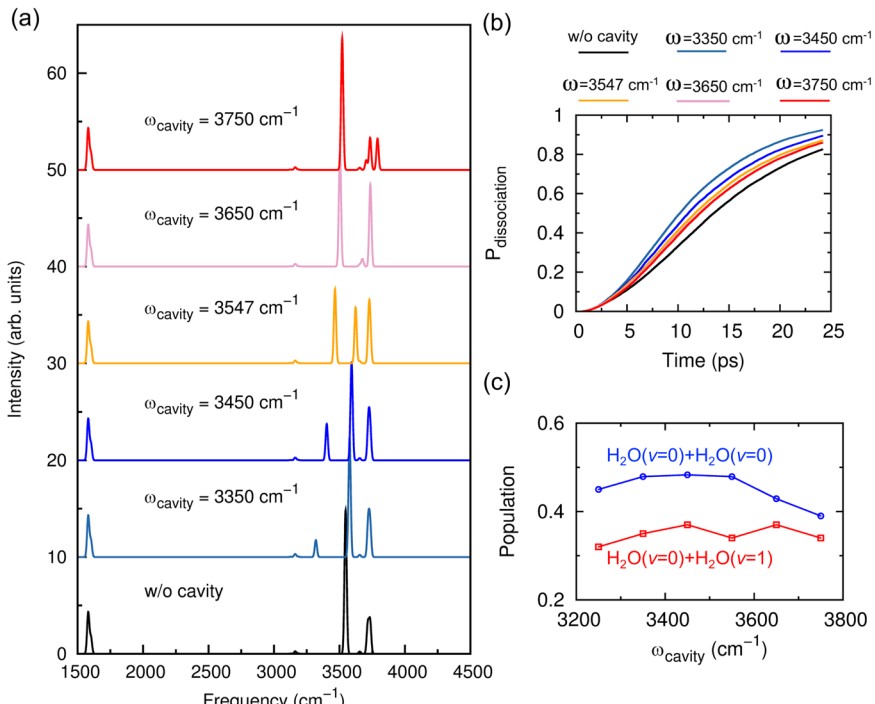

**Fig. 4 | Cavity frequency dependence of vibrational spectra, dissociation rates and pathways. a** Infrared spectra of $(H_2O)_2$ with different cavity mode frequencies ($\omega_{cavity}$). **b** Percentage of dissociated trajectories ($P_{dissociation}$) as a function of time for each $(H_2O)_2$-cavity system with different cavity mode frequencies ($\omega$). **c** Vibrational product population for the dissociation of the water dimer inside an optical cavity with different cavity mode frequencies. The dissociation channel of ground vibrational state products is indicated in blue and the channel of one bend-excited monomer fragment is indicated in red. ($v = 0$) indicates the vibrational ground state and ($v = 1$) indicates the bend fundamental state of the dissociation product. The light–matter coupling factors $g$ is set as 0.005 for **a**–**c**.

strength is modified. This indicates the fact that the underlying mechanisms of the two dissociation channels are similar under different VSC conditions upon $(H_2O)_2$ is inside the cavity.

**Effects of cavity frequency on reaction rates and pathways**

Thus far, we have demonstrated that both hydrogen-bond dissociation rate and channel can be significantly altered by the vibrational strong coupling in the $(H_2O)_2$-cavity system, with the resonant frequency of 3547 cm$^{-1}$ and different light–matter coupling strengths. Next, we investigate how the cavity mode detunings influence the spectral and dynamical properties of the $(H_2O)_2$-cavity system. Figure 4a displays the cav-VSCF/VCI spectra of $(H_2O)_2$-cavity system with different cavity mode frequencies. Together with the detailed spectral peak position, VCI coefficients, and assignments in Supplementary Table 5, it is seen that when the cavity frequency is lower than the HB stretch ($\omega_{cavity} = 3350$ or 3450 cm$^{-1}$), there still exists an LP state dominated by the cavity mode, while significant coupling with HB stretch, free OH stretch, and the bend overtone is also observed. On the contrary, the UP state is dominated by the HB stretch with couplings from the cavity modes and bend overtones. Since the HB stretch dominates the UP state, the corresponding IR intensity of the UP state is apparently higher than that of the LP state. When the cavity frequency is higher than the HB stretch ($\omega_{cavity} = 3650$ and 3750 cm$^{-1}$), the LP state is dominated by the HB stretch and the UP state is dominated by the cavity mode. Then the spectral peak of the LP state becomes more intense than the UP state. Strong mixing with the free OH stretch is also observed in both LP and UP states. However, as the cavity frequency increases, the bend overtone is found to have less contribution to the UP state.

Figure 4b shows the percentage of dissociated trajectories as a function of time for the cases with different cavity frequencies. First, upon $(H_2O)_2$ inside the cavity with coupling strength factor $g = 0.005$, the dissociation rate gets improved for all the cases. As we discussed before, under strong light–matter coupling, frequent energy exchange between the cavity field and $(H_2O)_2$ weakens the role of the cavity as energy reservoir. In addition, the existence of cavity induces the strong mode couplings involving either intramolecular OH stretches (like free OH stretch) or water bend overtone. Thus vibrational energy redistribution within $(H_2O)_2$ gets more efficient and accelerates the dissociation dynamics. Another interesting finding is that the hydrogen bond dissociation rate increases with smaller cavity frequencies. As indicated in the IR spectral results and Supplementary Table 5, with smaller cavity frequency, the coupling between HB stretch/cavity mode with the free OH stretch gets weaker while the coupling with bend overtone is more significant. Thus, the energy is easier to directly decay to the excitation of dissociation coordination and the bending mode of a water fragment. Finally, in Fig. 4c, we present the change of vibrational product populations with different cavity frequencies. In all cases, the major product corresponds to (000)+(000), which agrees with previous observations shown in Fig. 3. When the cavity frequency increases, the product distributions do not change dramatically. There seems to be a resonance effect on the population of (000)+(000) product. However, such effect is not apparent in the current work and needs more extensive investigations which are subject to our future studies. When the cavity frequency is larger than the resonance frequency (3547 cm$^{-1}$), both (000)+(000) and (000)+(010) channel populations tend to decrease. This can be attributed to the stronger coupling between HB stretch/cavity mode with intramolecular OH stretch, such as free OH stretch. As indicated in Supplementary Table 8, The corresponding dissociation channel could result in (000)+(100) or (000)+(001) with excited OH stretch in one water fragment. Note, such dissociation channels are energetically close quantum mechanically when $(H_2O)_2$ is outside the cavity. With the contribution from the cavity field, these channels can become minor dissociation channels.

## Discussion

To summarize, in this work, we present a novel combination of fully quantum cav-VSCF/VCI and quasi-classical trajectory approaches to investigate both spectral and dynamical properties of the water dimer when it is inside an optical cavity. The quantum dynamical IR spectra calculations provide direct evidence of how the cavity mode interacts with molecules and induces various vibrational mode couplings. Follow-up QCT calculations successfully reveal that the hydrogen bond dissociation dynamics in $(H_2O)_2$ can be significantly affected by the existence of an optical cavity. Besides the influence on intramolecular/intermolecular vibrational energy transfer, there exists competing effect on energy absorption and release for the cavity field. When the polaritonic vibrational coupling strength is relatively weak, the cavity field behaves as a reservoir of energy and inhibits the dissociation dynamics. With stronger light–matter coupling strength, the energy exchange between the molecule and cavity field is more frequent and the vibrational energy transfer process becomes more efficient, which in turn accelerates the reaction rate. One notable finding in our calculations is that the vibrational dissociation channels can be greatly influenced by the cavity. Upon $(H_2O)_2$ inside the cavity, with either weak or strong light–matter coupling, the ground state dissociation products can become the major products. Possible explanations involve (1) instead of inefficient decaying to the fundamental band, the system with excited bend overtone state can directly decay to the ground state with energy stored in the cavity field (2) a totally new vibrational dissociation channel where the energy transfer directly decays to excitation of the dissociative degree of freedom with excess energies absorbed by the cavity field. Both ways generates rotationally cold products. We also investigate the dependence of cavity mode frequencies on spectral and dynamical properties. It is confirmed that different cavity property could induce different extents of intramolecular/intermolecular vibrational energy transfer and affect the final dissociation dynamics. To obtain more quantitative understanding of the mechanism of the dissociation dynamics involving the vibrational relaxation and energy transfer processes, more rigorous theoretical approach, beyond the scope of quasi-classical trajectory method, should be applied. We previously investigated the vibrational relaxation pathways among water stretches and bending overtones in ice using the VSCF/VCI approach[54]. However, it still remains a challenge for related theoretical study of dissociation dynamics. We hope our work could stimulate further theoretical investigations in the future.

To the best of our knowledge, our work is the first study confirming the change of reactions in both reaction rate and pathways through accurate and statistically significant theoretical calculations, based on fully quantum and quasi-classical approaches and CCSD(T)-level PES/DMS. Although the current work focuses on gas-phase water dimer within a cavity and the realistic cavity system requires the existence of many molecules in a collective coupling regime, as well as the consideration of cavity loss, our work presents a solid first step for investigating the reaction dynamics of molecular systems in an optical cavity. The methodologies utilized in this work can be further applied to larger molecular systems from gas-phase clusters, i.e., $(H_2O)_{20}$, to condensed phase systems, i.e., liquid water and ice. Similar approaches have been applied to investigate the IR spectra and vibrational relaxation dynamics of liquid water and ice[43,54,55]. Future investigations will focus on understanding the collective spectral properties and dynamical reactivities by putting many molecules in the cavity[21,56]. We also note that recently, new experimental platform for vibrational strong coupling in gas phase molecules is realized[57]. We anticipate that the findings in our high-quality calculations can further guide the experimentalists to investigate how the ubiquitous hydrogen bond dissociation dynamics can be manipulated by the polaritonic vibrational strong coupling.

## Methods

### Cavity QED Hamiltonian

The widely used Pauli-Fierz Hamiltonian[16,58,59] for the cavity quantum electrodynamics (QED) is employed throughout this work:

$$\hat{H}_{QED} = \hat{H}_M + \hat{H}_C + \hat{H}_{CM} \tag{1}$$

The three terms on the right side of Eq. (1) are the Hamiltonian for the molecule, cavity phonon field Hamiltonian, and the light–matter interaction term, respectively:

$$\hat{H}_M = \hat{T}_N(\mathbf{R}) + V(\mathbf{R}) \tag{2}$$

$$\hat{H}_C = \sum_k^{N_C} \frac{1}{2}\left(\hat{p}_k^2 + \omega_k^2 \hat{q}_k^2\right) \tag{3}$$

$$\hat{H}_{CM} = \sum_k^{N_C} \omega_k \sqrt{\frac{1}{\epsilon_0 \tilde{V}}} \hat{q}_k(\hat{\boldsymbol{\mu}} \cdot \mathbf{e}_k) + \frac{1}{2}\frac{1}{\epsilon_0 \tilde{V}}(\hat{\boldsymbol{\mu}} \cdot \mathbf{e}_k)^2 \tag{4}$$

In Eqs. (2)–(4), $\hat{T}_N(\mathbf{R})$ is the kinetic energy operator and $V(\mathbf{R})$ is the potential energy of the molecular system. $N_C$ is the number of cavity modes, and $\omega_k, \hat{p}_k$, and $\hat{q}_k$ are the frequency, momentum operator, and position operator, respectively, for the cavity mode $k$. $\hat{\boldsymbol{\mu}}$ is the dipole moment vector operator for the molecular system, $\epsilon_0$ is the permittivity, $\tilde{V}$ is the effective cavity volume, and $\mathbf{e}_k$ is the polarization vector of the field for the cavity mode $k$. In Eqs. (3) and (4), each cavity mode $k$ is assumed to have frequency $\omega_k$ and polarization vector $\mathbf{e}_k$. Moreover, the momentum operator $\hat{p}_k$ and position operator $\hat{q}_k$ can be expressed in terms of the photon mode creation and annihilation operators, $\hat{a}_k^\dagger$ and $\hat{a}_k$:

$$\hat{q}_k = \sqrt{\frac{\hbar}{2\omega_k}}\left(\hat{a}_k^\dagger + \hat{a}_k\right), \hat{p}_k = i\sqrt{\frac{\hbar\omega_k}{2}}\left(\hat{a}_k^\dagger - \hat{a}_k\right) \tag{5}$$

With detailed expressions of three terms in Eqs. (2)–(4), the Pauli-Fierz Hamiltonian for the molecular system within an optical cavity can be rewritten as:

$$\hat{H} = \hat{T}_N(\mathbf{R}) + V(\mathbf{R}) + \sum_k^{N_C}\left[\frac{1}{2}\hat{p}_k^2 + \frac{1}{2}\omega_k^2\left(\hat{q}_k + \sqrt{\frac{2}{\omega_k^3}}g\hat{\boldsymbol{\mu}} \cdot \mathbf{e}_k\right)^2\right] \tag{6}$$

with the light–matter coupling factor defined as $g = \sqrt{\omega_k/(2\epsilon_0 \tilde{V})}$. Here, we denote the effective potential $V_{eff}$ and total field energy $E_{field}$ as:

$$V_{eff}(\mathbf{R},\mathbf{q}) = V(\mathbf{R}) + \sum_k^{N_C}\frac{1}{2}\omega_k^2\left(\hat{q}_k + \sqrt{\frac{2}{\omega_k^3}}g\hat{\boldsymbol{\mu}} \cdot \mathbf{e}_k\right)^2 \tag{7}$$

$$E_{field} = \sum_k^{N_C}\left[\frac{1}{2}\hat{p}_k^2 + \frac{1}{2}\omega_k^2\left(\hat{q}_k + \sqrt{\frac{2}{\omega_k^3}}g\hat{\boldsymbol{\mu}} \cdot \mathbf{e}_k\right)^2\right] \tag{8}$$

The energy of the cavity modes $E_{cav}$ is expressed straightforwardly from the cavity photon filed Hamiltonian in Eq. (3). As seen in Eqs. (7) and (8), the effective potential and the field energy are dependent on the molecular orientation as a result of the interaction term between the molecular dipole moment vector and the polarization vector of the cavity mode. Since the additive contributions from cavity field and light–matter interaction in Eq. (7) can always be optimized to zero, the ground state equilibrium geometry as well as the binding energy of water dimer will not be altered by the cavity[60].

## Potential energy and dipole moment surfaces of water dimer

To conduct fully quantum vibrational spectra and quasi-classical trajectory simulations of realistic molecules in an optical cavity, high-accurate potential energy and dipole moment surfaces (PES and DMS) are required. Here, for the water dimer system, we employed CCSD(T) level PES and DMS within the scheme of many-body expansion. Specifically, the PES is taken from our recently reported CCSD(T) water potential, q-AQUA[37], with the following expression:

$$V_{(H_2O)_2} = \sum_{i=1}^{2} V_{w_i}^{(1)} + V_{w_1,w_2}^{(2)} \tag{9}$$

where $V_{w_i}^{(1)}$ is the 1-body water monomer potential from a spectroscopically accurate PES developed by Partridge et al.[61] $V_{w_1,w_2}^{(2)}$ is the 2-body interaction potential between two water molecules[37]. This 2-body potential was fitted based on 71,892 CCSD(T)/CBS 2-b energies using permutationally invariant polynomial (PIP) approach[62]. The dipole moment of the water dimer system is also represented in a many-body expansion:

$$\mu_{(H_2O)_2} = \sum_{i=1}^{2} \mu_{w_i}^{(1)} + \mu_{w_1,w_2}^{(2)} \tag{10}$$

The 1-body water dipole, $\mu_{w_i}^{(1)}$, is from a highly accurate DMS for $H_2O$ developed by Lodi et al.[63]. The 2-body dipole, $\mu_{w_1,w_2}^{(2)}$, is from the PIP fit to roughly 30,000 MP2/aVTZ dipole moment data[64]. More details of the PES and DMS of water dimer are referred to the q-AQUA water potential[37] and WHBB water DMS[64].

## cav-VSCF/VCI vibrational spectra calculations

To perform quantum simulation of the vibrational spectra of water dimer within an optical cavity, we applied our recently developed cavity vibrational self-consistent field/virtual state configuration interaction (cav-VSCF/VCI) approach[36]. The cav-VSCF/VCI approach applies the cavity QED Hamiltonian, expressed in Eq. (6), and uses the popular Watson Hamiltonian in normal mode representation for the molecular Hamiltonian. Similar to the conventional VSCF/VCI approach, the cav-VSCF/VCI method expands the hybrid nuclear-cavity vibrational wavefunction on the basis of VSCF states:

$$\Psi(\mathbf{Q},\mathbf{q}) = \sum_m C_m \Phi_m(\mathbf{Q},\mathbf{q}) = \sum_m C_m \prod_{i=1}^{N} \phi_i^m(Q_i) \prod_{i=1}^{N_C} \chi_i^m(q_i) \tag{11}$$

where $\Phi_m(\mathbf{Q},\mathbf{q})$ are the ground and excited states from the VSCF solutions which is represented as a direct product of $N$ molecular mode wavefunctions and $N_C$ cavity mode wavefunctions. The combination coefficients $C_m$ are determined by diagonalizing the VCI Hamiltonian matrix. Interested readers are referred to our recent paper for more details[36]. All the cav-VSCF/VCI calculations were performed through a modified version of MULTIMODE[65], with an interface to the water dimer PES and DMS as described above. We considered one cavity mode with polarization direction along water dimer's O-O axis, and all the bending and OH stretch modes of the water dimer. Hence, there are 6 molecular normal modes and 1 cavity mode included in the cav-VSCF/VCI calculations. A 4-mode representation (4MR) of the effective potential, $V_{eff}$, and a 3-mode representation (3MR) of the dipole moment were used. Such treatment has been successfully used in the quantum vibrational spectra calculations of neutral and protonated water clusters to achieve a good balance of accuracy and computational efficiency. Details of the $N$-mode representation are included in the

Supplementary Note 1. The VCI excitation space is (10,9,8,7) for single, double, triple, and quadruple excitations respectively, ensuring a sufficiently large VCI matrix.

## Quasi-classical trajectory calculations

The Quasi-classical trajectory calculations were performed to simulate the vibrational dissociation dynamics of $(H_2O)_2$ within an optical cavity. Based on the cavity QED Hamiltonian in Eq. (6), the equations of motion for both molecular and cavity degrees of freedom were derived with the Hamilton's equations and propagated using the velocity-verlet integration method. The standard normal mode sampling was applied to prepare the initial quasi-classical states by assigning harmonic zero-point energy (ZPE) to each molecular normal mode and an extra quanta of excitation to the H-bonded OH stretch fundamental. The total rotational angular momentum of the water dimer was set to zero. One cavity mode was considered with its polarization direction along the O-O axis and initialized with its harmonic ZPE. For each set of initial conditions of the optical cavity, such as cavity frequency and light–matter coupling factor, a total of 20,000 trajectories were propagated for about 25 ps each with a 0.12 fs time step. The final products were analyzed for dissociated trajectories with O-O distance larger than 10 Å. The vibrational and rotational states of the dissociated water fragments were determined in the standard way[42] and are described below.

## Product-state analysis

For dissociated $(H_2O)_2$ from the trajectories, we conducted product analysis over each $H_2O$ fragment. First, the total classical rovibrational energy of $H_2O$ was determined. Then the total rotational angular momentum was removed using the same method as in preparing initial conditions for QCT calculations. With zero angular momentum, the vibrational energy of $H_2O$ was calculated. The classical rotational energy was calculated as the difference between original rovibrational energy and the pure vibrational energy.

Next, we describe how the quantum vibrational state levels of $H_2O$ were determined. Taking the global minimum geometry of $H_2O$ as the reference geometry, normal mode analysis was conducted, which provides harmonic frequencies of bend, symmetric stretch, and asymmetric stretch, as well as the orthogonal matrix to transform the mass-scaled Cartesian coordinates to normal mode coordinates. Using this orthogonal matrix, as well as the Cartesian coordinates and velocities of product $H_2O$, the normal mode coordinates, $\mathbf{Q}$, and momenta, $\mathbf{P}$ were obtained by linear transformation. The harmonic vibrational energy for each normal mode is then calculated through corresponding transformed normal mode coordinate, $Q_i$, and momentum, $P_i$.

Finally, the non-integer harmonic action number, $n_i'$, is determined by applying the energy formula of quantum harmonic oscillator. The integer harmonic action number $n_i$ is simply obtained by rounding $n_i'$ to its nearest integer value. The final vibrational state of a product $H_2O$ is denoted as $\mathbf{n} = (n_1, n_2, n_3)$, where $n_1, n_2, n_3$ represents the action numbers of symmetric stretch, water bend, and asymmetric stretch respectively.

## Gaussian binning analysis

To obtain the final rovibrational distributions of the fragment products, we applied the Gaussian binning procedure (1GB)[42,53,66–68] where one Gaussian weight for each fragment was calculated according to its total vibrational energy. This procedure assigns small weights for trajectories that have fragments violating the ZPE and effectively addresses the well-known ZPE leaking issue of the QCT method. The approaches of determining the quantum vibrational state and the 1GB approach have been successfully verified in the previous studies of vibrational dissociation dynamics of the water dimer[42,53]. Here, we briefly describe the Gaussian binning procedure to get statistical

probability of certain vibrational state $\mathbf{n}$. Suppose there are $N(\mathbf{n})$ number of products in the specific state $\mathbf{n}$ from $N_{\text{traj}}$ QCT trajectories, the probability of state $\mathbf{n}$ is expressed as:

$$P_{\text{GB}}(\mathbf{n}) = \frac{\sum_i^{N(\mathbf{n})} G_i(\mathbf{n})}{\sum_{\mathbf{n}} \sum_i^{N(\mathbf{n})} G_i(\mathbf{n})} \tag{12}$$

where $G_i(\mathbf{n})$ is the Gaussian weight assigned to each product, $i$, for state $\mathbf{n}$. In the 1GB procedure[53,66–68], we applied one Gaussian weight function to each product. The weight is determined through the following expression:

$$G_i(\mathbf{n}) = \frac{\beta}{\sqrt{\pi}} \exp\left[-\beta^2 \left(\frac{E(\mathbf{n}_i') - E(\mathbf{n})}{2E(\mathbf{0})}\right)^2\right] \tag{13}$$

where $\beta = 2\sqrt{\ln 2}/\delta$ with $\delta$ as the FWHM, $E(\mathbf{0})$ is the harmonic ZPE, $E(\mathbf{n})$ is the harmonic vibrational energy of the product for state $\mathbf{n}$. Finally, $E(\mathbf{n}_i')$ is calculated as the sum of kinetic energy and anharmonic potential energy of the product using its Cartesian coordinates and velocities after ruling out the rotational angular momentum. It should be noted that in the current work, the weight of the correlated vibrational state of the two $H_2O$ products is calculated as the product of two Gaussians, such that,

$$G(n_1, n_2, n_3, n_4, n_5, n_6) = G(n_1, n_2, n_3) \times G(n_4, n_5, n_6) \tag{14}$$

## Data availability
The source data as well as the output files generated in this study are available on the public repository https://github.com/qiyuchem/Cavity-water-dimer.

## Code availability
The source codes for cav-VSCF/VCI, QCT and ML PES/DMS are available upon request to the authors.

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

## Acknowledgements

Q.Y. and J.M.B. thank the ARO, DURIP grant (W911NF-14-1-0471), for funding a computer cluster where most of the calculations were performed and current financial support from NASA (80NSSC20K0360).

## Author contributions

Q.Y. conceived the project, performed calculations and analyzed the data. J.M.B. supervised the research. All the authors contributed to write the manuscript.

## Competing interests

The authors declare no competing interests.
