## [Peer Review File · Nature Communications]

Manipulating hydrogen bond dissociation rates and mechanisms in water dimer through vibrational strong couplingREVIEWER COMMENTS

Reviewer #1 (Remarks to the Author):

The authors investigate the dissociation dynamics of the water dimer considering vibrational strong coupling between molecular vibrational and cavity photon modes using state-of-the-art theoretical methods. This work deserves to be published in Nature Communications, because (1) the applied methodology is quite novel, (2) the present study connects to a hot topic (polaritonic chemistry) and may open new research directions, and (3) the new results are very interesting and certainly will attract wide attention.

The authors should address the following minor comments before acceptance:

At "CCSD(T)-F12/aVTZ" F12a or F12b should be specified.

The meaning of / is unclear in Table 1 and Supplementary Tables 2-4.

The abbreviation NVE should be defined.

The authors denote the bending-excited state of H₂O as (100), whereas traditionally (010) would be the notation. Also in Table 2 and at some other places as well the notations are not standard (see, for example, ref. 47). These should be clarified.

"we applied the Gaussian binning procedure (1GB)^{42,47}"

The original reference for 1GB is J. Chem. Phys. 131, 244302 (2009), which should be added.

Additional credit should also be given to: J. Chem. Phys. 133, 164108 (2010) and J. Phys. Chem. A 116, 7467 (2012).

At the sentence "Interested readers are also referred to previous papers.⁴⁷" more than one citation would be desired.

At "Supplementary Note 3. Gaussian binning analysis" proper references about the 1GB approach should be added (see above).

Supplementary Table 2 needs to be reformatted (see last column).

In Supplementary Table 3 "Cavity mode frequency" labels seem to be odd at the lower parts of the table. I think they should be "Coupling factor g".

In Supplementary Figure 2 caption a closing parenthesis is missing at "(ν_a, J_a ".

In Supplementary Table 6 "g" should be deleted at "Cavity frequency g".

In Supplementary Table 7 a closing parenthesis after 1 is missing at the head of the table.

Reviewer #2 (Remarks to the Author):

The authors presented high-level chemical dynamical calculations on a single molecule strongly coupled to the cavity mode. The result unambiguously showed that strong coupling could modify the energy distribution and chemical reaction dynamics. The results are solid and have provided potential physical pictures of how strong coupling could influence chemical reactions at a single-molecule strong coupling limits, beyond the simple picture of "potential energy surface was modified" that was often presented in the literature. However, I do have a few important comments for the authors to consider

before this manuscript is ready to be published in Nature Communication.

First and foremost, the authors should emphasize more that their results are at the single-molecule level, which is many orders of magnitudes away from the experimental conditions (with the number of molecules to be 10^6 to 10^{10} in the cavity). I applaud that the authors did not dodge this point and rightly mentioned it on page 6. I think it is useful for the author to be specific about how small the cavity volume needs to be, is it a nanocavity or picocavity? This is critical. There are so many theoretical papers performing strong coupling at the single-molecule level and claiming that they clearly explained why strong coupling could modify chemical reactions in the collective coupling regime in experiments. These claims are no more than alternative facts. The truth is that in the experiment, the coupling strength is shared by 10^{10} molecules, while in the simulation, the coupling strength is between the cavity and a single molecule. How could the single-molecule strong coupling case to be equivalent to the collective strong coupling at all? I want to point out that this is an issue of the community instead of the authors themselves. Thus, I still believe the present work is of high importance, but I also believe that if the authors can clearly articulate the difference between their results and experimental reality, it won't weaken the impact of their work and instead will serve the community by pointing out the gap and future directions.

second, in page 20, the authors claimed, "our work lays a solid foundation for investigating the reaction dynamics of molecular systems in an optical cavity." this is quite an overstatement for the exact reason I mentioned in the paragraph above. In my opinion, a solid foundation will be established if theoreticians can investigate the reactions under collective coupling effects, or experimentalists can study reaction dynamics under single molecule strong coupling conditions, or at least be able to state-resolve dynamics of polariton versus dark modes. What the authors did would be a solid step towards to the foundation of understanding reaction dynamics under collective strong coupling.

third, page 15, "such process can be realized due to the existence of cavity mode and its coupling with the bend overtone". It was unclear to me how the bend overtone coupled with the cavity mode as its own oscillator strength is small. is it due to intrinsic intramolecular coupling (i.e., Fermi resonance)? for the same sentence, the authors try to give an explanation about the population change by proposing alternative dynamic channels, such as relaxing the bend overtone through photonic coupling. It is important that the authors be more quantitative instead of speculative here. for example, what is the timescale of bend overtone relaxation? Is it even competitive to relax to bend fundamentals?

fourth, page 16, similarly, "one reason is that ... leaving the fragment in its bend overtone of PH stretch excited state". The explanation here is rather speculative instead of being affirmative and quantitative. Based on the simulation results, can the authors quantify the relative population in the dissociation degree of freedom? this would make the argument strong and convincing. There are a few other places where the statement was also rather speculative, and it would be great if the authors can make it more quantitative.

overall, I found this work of high quality, but the authors should address the two overall concerns: 1. clearly explain the difference between their single molecule strong coupling simulation and the collective strong coupling experiments. 2. be quantitative as possible as you could and reduce the number of speculative statements.

Reviewer #3 (Remarks to the Author):

This manuscript from Yu and Bowman demonstrates the effect of vibrational strong coupling on the dissociation of a water dimer. The authors thoroughly examined the effects of coupling strength, as well as cavity frequency, on both the reaction rate and dissociation mechanism. I found this work to be very exciting, and would recommend its publication after the authors address a few minor technical

points.

- In the field of polariton chemistry, Jaynes-Cummings model is heavily used. It might be helpful to point out whether the vibrational frequencies in Table 1 can be obtained from a simple Jaynes-Cummings model, where 6 normal modes are coupled to the cavity through dipole-field interactions.

- Can the nonsymmetric Rabi splitting in Fig. 1c (in terms of the intensities of the lower and upper polaritons) be explained using simple perturbation theory (similar to those in Appendix C of 10.1063/5.0057542)?

- It seems that the gas-phase dimer geometry was used to acquire the vibrational frequencies in Fig. 1c and Table 1. Does a change in the vibrational frequencies suggest a (small) change to the local curvatures of the PES? Or the local curvatures are preserved, but an extra degree of freedom (cavity) causes the changes in the frequencies? Relatedly, does the cavity modify the ground-state equilibrium geometry and the binding energy of two water molecules?

- In Figure 1b, is Q (the x-axis) the mass-weighted normal coordinate for the H-bonded OH stretch? The values (-20 to 30 a.u.) seem a little too large.

- Can the populations for different rotational levels in Figure 3b-d be integrated to get the total populations in Figure 3a?

- Can the authors point out whether the advanced methodologies utilized in the work are applicable to large molecules or many molecules in the cavity?

Response to Reviews

We thank the three reviewers for their helpful comments and suggestions. We have revised the paper in response to these comments. The original comments from the reviewers are in black font. Our point-by-point responses to the comments are in blue font. Changes in the manuscript are indicated in red font and are highlighted in the marked-up version of the revised manuscript.

Reviewer #1:

The authors investigate the dissociation dynamics of the water dimer considering vibrational strong coupling between molecular vibrational and cavity photon modes using state-of-the-art theoretical methods. This work deserves to be published in Nature Communications, because (1) the applied methodology is quite novel, (2) the present study connects to a hot topic (polaritonic chemistry) and may open new research directions, and (3) the new results are very interesting and certainly will attract wide attention.

The authors should address the following minor comments before acceptance:

We thank the reviewer for the predominantly favorable assessment of our work and the helpful comments. We address the specific comments below.

At "CCSD(T)-F12/aVTZ" F12a or F12b should be specified.

We used "CCSD(T)-F12b/aVTZ". This has been specified in the manuscript.

The meaning of / is unclear in Table 1 and Supplementary Tables 2-4.

We thank the reviewer for pointing this out. "/" means the contribution from corresponding mode is almost neglectable with VCI coefficient smaller than 0.005. We have clarified this point in the revised Table 1 and Supplementary Tables.

The abbreviation NVE should be defined.

We have added the microcanonical ensemble definition of NVE in the manuscript:

"where the sum of E_{mol} and E_{field} is set as a constant along the microcanonical (NVE) trajectory."

The authors denote the bending-excited state of H₂O as (100), whereas traditionally (010) would be the notation. Also in Table 2 and at some other places as well the notations are not standard (see, for example, ref. 47). These should be clarified.

We thank the reviewer for pointing this out. We have switched (100) to (010) to indicate the bending-excited state throughout the text and also added detailed clarifications in Table 2.

"we applied the Gaussian binning procedure (1GB)^{42,47}"

The original reference for 1GB is J. Chem. Phys. 131, 244302 (2009), which should be added. Additional credit should also be given to: J. Chem. Phys. 133, 164108 (2010) and J. Phys. Chem. A 116, 7467 (2012).

We thank the reviewer a lot for pointing out these important references. We have added them when introducing the Gaussian binning procedure.

At the sentence "Interested readers are also referred to previous papers.⁴⁷" more than one citation would be desired.

Done!

At "Supplementary Note 3. Gaussian binning analysis" proper references about the 1GB approach should be added (see above).

Done!

Supplementary Table 2 needs to be reformatted (see last column).

Done!

In Supplementary Table 3 "Cavity mode frequency" labels seem to be odd at the lower parts of the table. I think they should be "Coupling factor g".

We thank the reviewer for pointing this out. These have been fixed.

In Supplementary Figure 2 caption a closing parenthesis is missing at "(va,Ja".

Fixed!

In Supplementary Table 6 "g" should be deleted at "Cavity frequency g".

Fixed!

In Supplementary Table 7 a closing parenthesis after 1 is missing at the head of the table.

Fixed!

Reviewer #2:

The authors presented high-level chemical dynamical calculations on a single molecule strongly coupled to the cavity mode. The result unambiguously showed that strong coupling could modify the energy distribution and chemical reaction dynamics. The results are solid and have provided potential physical pictures of how strong coupling could influence chemical reactions at a single-molecule strong coupling limits, beyond the simple picture of "potential energy surface was modified" that was often presented in the literature. However, I do have a few important comments for the authors to consider before this manuscript is ready to be published in Nature Communication.

We thank the reviewer for the predominantly favorable assessment of our work and the helpful comments. We address the specific comments below.

First and foremost, the authors should emphasize more that their results are at the single-molecule level, which is many orders of magnitudes away from the experimental conditions (with the number of molecules to be 10^6 to 10^{10} in the cavity). I applaud that the authors did not dodge this point and rightly mentioned it on page 6. I think it is useful for the author to be specific about how small the cavity volume needs to be, is it a nanocavity or picocavity? This is critical. There are so many theoretical papers performing strong coupling at the single-molecule level and claiming that they clearly explained why strong coupling could modify chemical reactions in the collective coupling regime in experiments. These claims are no more than alternative facts. The truth is that in the experiment, the coupling strength is shared by 10^{10} molecules, while in the simulation, the coupling strength is between the cavity and a single molecule. How could the single-molecule strong coupling case to be equivalent to the collective strong coupling at all? I want to point out that this is an issue of the community instead of the authors themselves. Thus, I still believe the present work is of high importance, but I also believe that if the authors can clearly articulate the difference between their results and experimental reality, it won't weaken the impact of their work and instead will serve the community by pointing out the gap and future directions.

We appreciate the reviewer pointing out the potential issue of single molecule simulations. We strongly agree with the reviewer that it is important to indicate the difference between the single molecule system in the current work and the experimental conditions. We have rewritten and expanded the discussion about this point on Page 6:

“It should be noted that in the vibrational strong coupling experiments, a macroscopic number of molecules, N_{mol} (i.e. 10^{10}), are collectively coupled to the cavity and the resulting Rabi splitting is proportional to $\sqrt{N_{\text{mol}}/\tilde{V}}$. In the current work, the light-matter coupling is directly between the cavity and the water dimer. Thus, to compensate for the lack of N_{mol} molecules in our $(\text{H}_2\text{O})_2$ -cavity system, the coupling strength factor g throughout this work indicates a much smaller cavity volume \tilde{V} than the experimental set-up. For example, with cavity frequency $\omega=3547 \text{ cm}^{-1}$ and coupling strength factor $g=0.005 \text{ a.u.}$, the effective cavity volume \tilde{V} calculated from Equation 4 is 0.6 nm^3 which is within the picocavity range. We also emphasize that the single-molecule simulations in this work serve as a good example to investigate how the optical cavity affects the vibrational mode couplings and chemical reaction dynamics. To bridge the gap between single-molecule simulation and realistic experimental conditions in collective coupling regime, the many-

molecule system will need to be considered and this is subject to our future work using extensions of the approaches as described in Methods section.”

second, in page 20, the authors claimed, "our work lays a solid foundation for investigating the reaction dynamics of molecular systems in an optical cavity." this is quite an overstatement for the exact reason I mentioned in the paragraph above. In my opinion, a solid foundation will be established if theoreticians can investigate the reactions under collective coupling effects, or experimentalists can study reaction dynamics under single molecule strong coupling conditions, or at least be able to state-resolve dynamics of polariton versus dark modes. What the authors did would be a solid step towards to the foundation of understanding reaction dynamics under collective strong coupling.

We thank the reviewer for pointing this out. As we replied in the last comment, we agree that it is important to investigate the chemical reactions affected by the optical cavity under collective coupling conditions. In the current work, we are only focusing on single-molecule (water dimer) reactions in an optical cavity, but it can be a solid first step toward the final goal. We are exploring the vibrational dynamics in more complicated systems from larger gas-phase water clusters, i.e., $(\text{H}_2\text{O})_{20}$, to condensed phase systems, i.e., liquid water and ice. We plan to report such work in the future and again, the water dimer within optical cavity in the current work is the first step. Additionally, our theoretical approaches can be naturally applied to reaction dynamics and vibrational spectral studies of systems with many molecules, using highly accurate and efficient machine learning potential and dipole moment surfaces. Previously, we have used similar approaches to simulate the IR spectra of liquid water and ice as well as the vibrational relaxation dynamics.[JCP,142,194502,2015; JPCL, 3,3671, 2012; JACS, 136, 5888, 2014]

We have modified the discussion of the impact of our work and indicated how our approaches can be applied to study the cavity systems under collective coupling regime:

“Although the current work focuses on gas-phase water dimer within a cavity and the realistic cavity system requires the existence of many molecules in a collective coupling regime, as well as the consideration of cavity loss, our work presents a solid first step for investigating the reaction dynamics of molecular systems in an optical cavity. The methodologies utilized in this work can be further applied to larger molecular systems from gas-phase clusters, i.e. $(\text{H}_2\text{O})_{20}$, to condensed phase systems, i.e. liquid water and ice. Similar approaches have been applied to investigate the IR spectra and vibrational relaxation dynamics of liquid water and ice.^{43,54,55}”

third, page 15, "such process can be realized due to the existence of cavity mode and its coupling with the bend overtone". It was unclear to me how the bend overtone coupled with the cavity mode as its own oscillator strength is small. is it due to intrinsic intramolecular coupling (i.e., Fermi resonance)? for the same sentence, the authors try to give an explanation about the population change by proposing alternative dynamic channels, such as relaxing the bend overtone through photonic coupling. It is important that the authors be more quantitative instead of speculative here. for example, what is the timescale of bend overtone relaxation? Is it even competitive to relax to bend fundamentals?

We thank the reviewer for the comment on how the bend overtone is involved in the reaction. From our cav-VSCF/VCI calculation (shown in Table 1), there exists direct couplings between the bend overtone and the HB stretch, as well as the cavity mode when the water dimer is within the cavity. The coupling between bend overtone and HB stretch was observed in various water systems, such as water hexamers [JPCL,4,1104,2013] and ice [JPCL,3,3671,2012]. In water dimer, their coupling also exists although it is relatively weak.

As for the coupling between bend overtone and cavity mode, this comes from the intermediate role of the HB stretch. We performed additional cav-VSCF/VCI calculations by including the water dimer in the cavity but without consideration of the HB stretch. As shown in Supplementary Figure 2 and Supplementary Table 3, the bend overtone does not couple with the cavity mode when the HB stretch is excluded. When the HB stretch is included, the couplings among bend overtone, HB stretch, and cavity mode are significant. These observations provide direct support that the energy transfer between bend overtone and cavity mode will be efficient and could affect the reaction channel for dissociation. Finally, the vibrational relaxation from bend overtone to fundamental is not efficient considering their very small coupling, as shown in Supplementary Table 3.

We also want to indicate that it is theoretically challenging to monitor the vibrational relaxation process and intramolecular vibrational redistribution during dynamics. Previously, we investigated the vibrational relaxation dynamics in ice using the VSCF/VCI approach.[JACS, 136, 5888, 2014] However, to the best of our knowledge, no rigorous theoretical efforts have been made in the dissociation dynamics. We hope our current work can stimulate related theoretical investigations.

We have rewritten the related discussions with support from new calculations:

“The significant modification of the dissociation channel by the cavity can be attributed to the novel couplings among HB stretch, bend overtone, and the cavity mode. After the bend overtone state (020) is excited through IVR process, direct decaying to the bend fundamental state (010) is not efficient due to their small vibrational couplings as indicated in Supplementary Table 3. Instead, the bend overtone can directly decay to the ground state (000) with energy absorbed by the cavity field. This is realized through HB stretch-mediated coupling between bend overtone and cavity mode. As shown in Supplementary Figure 2 and Supplementary Table 3 where we performed cav-VSCF/VCI calculation with and without the inclusion of HB stretch, the coupling between bend overtone and cavity mode becomes significant only when the HB stretch is included in the calculation. The fact that the HB stretch strongly couples with the cavity mode provides another new vibrational dissociation channel with the existence of VSC. Instead of transferring to the bend

overtone and decaying to one quanta of bending excitation in a fragment, the vibrational energy of the excited HB stretch can be directly absorbed by the cavity field due to their strong coupling.”

fourth, page 16, similarly, "one reason is that leaving the fragment in its bend overtone of PH stretch excited state". The explanation here is rather speculative instead of being affirmative and quantitative. Based on the simulation results, can the authors quantify the relative population in the dissociation degree of freedom? this would make the argument strong and convincing. There are a few other places where the statement was also rather speculative, and it would be great if the authors can make it more quantitative.

We thank the reviewer for this comment and suggestions. We have rewritten related discussions and pointed out the theoretical challenges in obtaining a quantitative picture of vibrational relaxation and energy transfer during dissociation dynamics:

“As discussed in the section of IR spectra and also in Table 1, when the light-matter coupling strength increases, the couplings among cavity, HB stretch, free OH stretch, symmetric OH stretch, and the bend overtone becomes more significant. Additionally, when the intermolecular low-frequency modes (i.e. O-O stretch) are considered, as shown in Supplementary Table 2, vibrational modes such as bend overtone and free OH stretch are further mixed with the cavity mode and HB stretch with larger VCI coefficients. Thus, the intramolecular and intermolecular energy transfer involving these modes becomes more efficient. The energy can decay to the excitation of dissociation degree of freedom which is related with the low-frequency modes, leaving the fragment in its bend overtone or OH stretch excited state.”

In the Discussion section:

“To obtain more quantitative understanding of the mechanism of the dissociation dynamics involving the vibrational relaxation and energy transfer processes, more rigorous theoretical approach, beyond the scope of quasi-classical trajectory method, should be applied. We previously investigated the vibrational relaxation pathways among water stretches and bending overtones in ice using the VSCF/VCI approach.⁵⁴ However, it still remains a challenge for related theoretical study of dissociation dynamics. We hope our work could stimulate further theoretical investigations in the future.”

overall, I found this work of high quality, but the authors should address the two overall concerns: 1. clearly explain the difference between their single molecule strong coupling simulation and the collective strong coupling experiments. 2. be quantitative as possible as you could and reduce the number of speculative statements.

We have addressed these concerns when addressing the specific comments (1)-(4) above. We hope the related discussions are clear and informative now.

Reviewer #3:

This manuscript from Yu and Bowman demonstrates the effect of vibrational strong coupling on the dissociation of a water dimer. The authors thoroughly examined the effects of coupling strength, as well as cavity frequency, on both the reaction rate and dissociation mechanism. I found this work to be very exciting, and would recommend its publication after the authors address a few minor technical points.

We thank the reviewer for the predominantly favorable assessment of our work and the helpful comments. We address the specific comments below.

- In the field of polariton chemistry, Jaynes-Cummings model is heavily used. It might be helpful to point out whether the vibrational frequencies in Table 1 can be obtained from a simple Jaynes-Cummings model, where 6 normal modes are coupled to the cavity through dipole-field interactions.

We thank the reviewer for pointing this out. The widely used Jaynes-Cummings model could be applied to obtain vibrational frequencies of polaritonic states by solving simplified polariton Hamiltonian. However, it has been shown that the Jaynes-Cummings model is not very accurate in describing rovibrational transitions (JPCL, 11, 7525-7530, 2020). Besides, such approach is limited to two-level systems and ignores the permanent dipole and dipole self-energy, which are important when the coupling strength increases. In our cav-VSC/VCI approach, we employed a rigorous Hamiltonian and could describe multi-level couplings among molecular vibrations and cavity modes.

We have added related discussions before introducing the cav-VSCF/VCI spectra:

“It should also be noted that, to obtain frequencies of polaritonic states, the ubiquitous Jaynes-Cummings model^{44,45} could be applied by solving simplified polariton Hamiltonian matrix. However, it has been shown that the Jaynes-Cummings model is not accurate in describing rovibrational transitions.⁴⁶ Such method is also limited to two-level systems and ignores the permanent dipole and dipole self-energy.⁴⁷ Our cav-VSCF/VCI approach applies a rigorous Hamiltonian and could accurately describe multi-level couplings among molecular vibrations and cavity modes.³⁷”

- Can the nonsymmetric Rabi splitting in Fig. 1c (in terms of the intensities of the lower and upper polaritons) be explained using simple perturbation theory (similar to those in Appendix C of 10.1063/5.0057542)?

We thank the reviewer for pointing this out. From the view of vibrational configurational interaction, the intensities of the lower and upper polariton are attributed to the amount of molecular vibrations in the hybrid polaritonic states (see corresponding VCI coefficients). Similar behavior could be realized by performing perturbation theory calculations. It would be interesting to test these approximate methods.

We have added related discussions:

“It can also be observed in Figure 1(c) that the Rabi splitting patterns are slightly asymmetric with increasing light-matter coupling strength. As seen in Table 1, the asymmetric intensities of LP and UP states are attributed to different amounts of molecular vibrational modes in the formation of hybrid polaritonic state (see corresponding VCI coefficients). It would also be interesting to test approximate methods, such as vibrational perturbation theory,^{50,51} to capture the asymmetric Rabi splitting features.”

- It seems that the gas-phase dimer geometry was used to acquire the vibrational frequencies in Fig. 1c and Table 1. Does a change in the vibrational frequencies suggest a (small) change to the local curvatures of the PES? Or the local curvatures are preserved, but an extra degree of freedom (cavity) causes the changes in the frequencies? Relatedly, does the cavity modify the ground-state equilibrium geometry and the binding energy of two water molecules?

We appreciate the reviewer’s comments on possible modifications to the local curvatures of the PES. With the extra degree of freedom of cavity modes, the curvatures of the multidimensional PES will be modified, which results in different coupling patterns among molecule’s vibrational modes and cavity modes, as well as the change of IR spectra.

For the ground-state equilibrium geometry and binding energy, from the effective potential V_{eff} in Equation (5), it can be inferred that the ground state equilibrium geometry is not changed since the potential energy contributions from the cavity field and light-matter interaction can always be optimized to zero. Due to the same reason, the binding energy of water dimer is not affected by the cavity.

We have added related discussions after Equation (5):

“Since the additive contributions from cavity field and light-matter interaction in Equation 5 can always be optimized to zero, the ground state equilibrium geometry as well as the binding energy of water dimer will not be altered by the cavity⁵⁸.”

- In Figure 1b, is Q (the x-axis) the mass-weighted normal coordinate for the H-bonded OH stretch? The values (-20 to 30 a.u.) seem a little too large.

The unit, Q , in Figure 1b is mass-weighted normal coordinates of H-bonded OH stretch. We used the range -20 to 30 a.u. to allow the total potential energy greater than 40 kcal/mol ($\sim 25000 \text{ cm}^{-1}$) which is adequate to include highly excited vibrational states in our cav-VSCF/VCI calculations.

We have modified related descriptions in the figure captions:

“Effective potential energy surfaces along the mass-weighted normal coordinates of hydrogen-bonded OH stretch Q and the cavity mode Q_c ”

- Can the populations for different rotational levels in Figure 3b-d be integrated to get the total populations in Figure 3a?

We thank the reviewer for pointing this out. Integration over the rotation levels for all products could be done straightforwardly. However, we do not really see strong motivations here because it will lose important information about the specific vibrational states. Also, the rotational distribution patterns in Figure 3b-d are similar and the reaction products are not highly rotational excited. We could anticipate that the total rotational distribution regardless of the vibrational states will also be similar to Figure 3b-d.

- Can the authors point out whether the advanced methodologies utilized in the work are applicable to large molecules or many molecules in the cavity?

We thank the reviewer for pointing this out. With accurate and efficient machine learning potential energy and dipole moment surfaces, our approach can be naturally extended to more complicated systems with large molecules or many molecules in the cavity. This is subject to our next research direction to investigate chemical reactions in the collective coupling regime.

We have added related discussions in the section of discussion:

“Although the current work focuses on gas-phase water dimer within a cavity and the realistic cavity system requires the existence of many molecules in a collective coupling regime, as well as the consideration of cavity loss, our work presents a solid first step for investigating the reaction dynamics of molecular systems in an optical cavity. The methodologies utilized in this work can be further applied to larger molecular systems from gas-phase clusters, i.e. $(\text{H}_2\text{O})_{20}$, to condensed phase systems, i.e. liquid water and ice. Similar approaches have been applied to investigate the IR spectra and vibrational relaxation dynamics of liquid water and ice.^{43,54,55}”

We thank the three reviewers again for their thoughtful and helpful comments. We hope that this paper is now suitable for publication in *Nature Communication*.

REVIEWERS' COMMENTS

Reviewer #2 (Remarks to the Author):

The authors have addressed my questions nicely. I recommend it to be published as-is.

Reviewer #3 (Remarks to the Author):

I appreciate the responses from the authors on my previous comments. The only very minor issue is that "vibrational perturbation theory,50,51" should be changed to "perturbation theory,50,51", because Ref. 51 focused on electronic instead of vibrational states. After this change, I would recommend the publication of this manuscript in Nature Communications.

Response to Reviews

We thank the reviewers again for their time and helpful comments. We have revised the paper in response to these comments. The original comments from the reviewers are in black font. Our point-by-point responses to the comments are in blue font. Changes in the manuscript are highlighted in the marked-up version of the revised manuscript.

Reviewer #2:

The authors have addressed my questions nicely. I recommend it to be published as-is.

We thank the reviewer for the recommendation of our work.

Reviewer #3:

I appreciate the responses from the authors on my previous comments. The only very minor issue is that "vibrational perturbation theory,50,51" should be changed to "perturbation theory,50,51", because Ref. 51 focused on electronic instead of vibrational states. After this change, I would recommend the publication of this manuscript in Nature Communications.

We thank the reviewer for pointing this out. We have revised the manuscript accordingly.